# Numerical Accounting in the Shuffle Model of Differential Privacy

**Antti Koskela**                                                    *antti.h.koskela@nokia-bell-labs.com*
*Nokia Bell Labs*
*University of Helsinki*

**Mikko Heikkilä**                                                    *mikko.a.heikkila@helsinki.fi*
*Department of Computer Science*
*University of Helsinki*

**Antti Honkela**                                                    *antti.honkela@helsinki.fi*
*Department of Computer Science*
*University of Helsinki*

**Reviewed on OpenReview:** *https://openreview.net/forum?id=11osftjEbF*

## Abstract

Shuffle model of differential privacy is a novel distributed privacy model based on a combination of local privacy mechanisms and a secure shuffler. It has been shown that the additional randomisation provided by the shuffler improves privacy bounds compared to the purely local mechanisms. Accounting tight bounds, however, is complicated by the complexity brought by the shuffler. The recently proposed numerical techniques for evaluating $(\varepsilon, \delta)$-differential privacy guarantees have been shown to give tighter bounds than commonly used methods for compositions of various complex mechanisms. In this paper, we show how to utilise these numerical accountants for adaptive compositions of general $\varepsilon$-LDP shufflers and for shufflers of $k$-randomised response mechanisms, including their subsampled variants. This is enabled by an approximation that speeds up the evaluation of the corresponding privacy loss distribution from $\mathcal{O}(n^2)$ to $\mathcal{O}(n)$, where $n$ is the number of users, without noticeable change in the resulting $\delta(\varepsilon)$-upper bounds. We also demonstrate looseness of the existing bounds and methods found in the literature, improving previous composition results for shufflers significantly.

## 1 Introduction

The shuffle model of differential privacy (DP) is a distributed privacy model which sits between the high trust–high utility centralised DP, and the low trust–low utility local DP (LDP). In the shuffle model, the individual results from local randomisers are only released through a secure shuffler. This additional randomisation leads to "amplification by shuffling", resulting in better privacy bounds against adversaries without access to the unshuffled local results.

We consider computing privacy bounds for both single and composite shuffle protocols, where by composite protocol we mean a protocol, where the subsequent user-wise local randomisers depend on the same local datasets and possibly on the previous output of the shuffler, and at each round the results from the local randomisers are independently shuffled. Moreover, using the analysis by Feldman et al. (2023), we provide bounds in the case the subsequent local randomisers are allowed to depend adaptively on the output of the previous ones.

In this paper we show how numerical accounting (Koskela et al., 2020; 2021; Gopi et al., 2021) can be employed for privacy analysis of both single and composite shuffle DP mechanisms. We demonstrate that

thus obtained bounds can be up to orders of magnitudes tighter than the existing bounds from the literature. We also evaluate how significantly adversaries with varying capabilities differ in terms of the resulting privacy bounds using the $k$-randomised response mechanism. For conciseness, most of the proofs are given in the Appendix.

## 1.1 Related work

DP was originally defined in the central model assuming a trusted aggregator by Dwork et al. (2006), while the fully distributed LDP was formally introduced and analysed by Kasiviswanathan et al. (2011). Closely related to the shuffle model of DP, Bittau et al. (2017) proposed the Encode, Shuffle, Analyze framework for distributed learning, which uses the idea of secure shuffler for enhancing privacy. The shuffle model of DP was formally defined by Cheu et al. (2019), who also provided the first separation result showing that the shuffle model is strictly between the central and the local models of DP. Another direction initiated by Cheu et al. (2019) and continued, e.g., by Balle et al. (2020b); Ghazi et al. (2021) has established a separation between single- and multi-message shuffle protocols.

There exists several papers on privacy amplification by shuffling, some of which are central to this paper. Erlingsson et al. (2019) showed that the introduction of a secure shuffler amplifies the privacy guarantees against an adversary, who is not able to access the outputs from the local randomisers but only sees the shuffled output. Balle et al. (2019) improved the amplification results and introduced the idea of privacy blanket, which we also utilise in our analysis of $k$-randomised response. Feldman et al. (2021) used a related idea of hiding in the crowd to improve on the previous results, and their analysis was further improved in (Feldman et al., 2023). Girgis et al. (2021) generalised shuffling amplification further to scenarios with composite protocols and parties with more than one local sample under simultaneous communication and privacy restrictions. We use the improved results of Feldman et al. (2023) in the analysis of general LDP mechanisms, and compare our bounds with theirs in Section 3.3. We also calculate privacy bounds in the setting considered by Girgis et al. (2021), namely in the case where a subset of users sending contributions to the shufflers are sampled randomly. This can be seen as a subsampled mechanism and we are able to combine the analysis of Feldman et al. (2023), the privacy loss distribution related subsampling results of Zhu et al. (2022) and FFT accounting to obtain tighter $(\varepsilon, \delta)$-bounds than Girgis et al. (2021), as shown in Section 3.4.

## 2 Background: numerical privacy accounting

Before analysing the shuffled mechanisms we introduce some required theory and notations. In particular, we use the privacy loss distribution formalism, which is based on finding the so-called dominating pairs of distributions for the given mechanisms. For more detailed presentations of the theory, we refer to Koskela et al. (2021); Gopi et al. (2021); Zhu et al. (2022).

### 2.1 Differential privacy and privacy loss distribution

An input dataset containing $n$ data points is denoted as $X = (x_1, \ldots, x_n) \in \mathcal{X}^n$, where $x_i \in \mathcal{X}$, $1 \le i \le n$. We say $X, X' \in \mathcal{X}^n$ are neighbours if we get one by substituting one element in the other (denoted $X \sim X'$).

**Definition 1.** *Let $\varepsilon > 0$ and $\delta \in [0, 1]$. Let $P$ and $Q$ be two random variables taking values in the same measurable space $\mathcal{O}$. We say that $P$ and $Q$ are $(\varepsilon, \delta)$-indistinguishable, denoted $P \simeq_{(\varepsilon,\delta)} Q$, if for every measurable set $E \subset \mathcal{O}$ we have*

$$\Pr(P \in E) \le e^\varepsilon \Pr(Q \in E) + \delta, \qquad \Pr(Q \in E) \le e^\varepsilon \Pr(P \in E) + \delta.$$

**Definition 2.** *Let $\varepsilon > 0$ and $\delta \in [0, 1]$. Mechanism $\mathcal{M} : \mathcal{X}^n \to \mathcal{O}$ is $(\varepsilon, \delta)$-DP if for every $X \sim X'$: $\mathcal{M}(X) \simeq_{(\varepsilon,\delta)} \mathcal{M}(X')$. We call $\mathcal{M}$ tightly $(\varepsilon, \delta)$-DP, if there does not exist $\delta' < \delta$ such that $\mathcal{M}$ is $(\varepsilon, \delta')$-DP.*

When the data are distributed among several parties, and the local datasets are only accessed via purely local DP mechanisms, we say that the mechanisms guarantee local DP (LDP) and call the local DP mechanisms local randomisers (Kasiviswanathan et al., 2011).

We rely on the results of Zhu et al. (2022) and characterise $(\varepsilon, \delta)$-DP bounds using the hockey-stick divergence, which for $\alpha \geq 0$ is defined as

$$H_\alpha(P||Q) = \int [P(t) - \alpha \cdot Q(t)]_+ \, \mathrm{d}t,$$

where for $x \in \mathbb{R}$, $x_+ = \max\{0, x\}$. Using the hockey-stick divergence, by (Lemma 5, Zhu et al., 2022), tight $(\varepsilon, \delta)$-DP bounds can also be characterised as

$$\delta(\varepsilon) = \max_{X \sim X'} H_{\mathrm{e}^\varepsilon}(\mathcal{M}(X)||\mathcal{M}(X')).$$

We can generally find $(\varepsilon, \delta)$-bounds by analysing dominating pairs of distributions:

**Definition 3** (Zhu et al. 2022). *A pair of distributions $(P, Q)$ is a dominating pair of distributions for mechanism $\mathcal{M}$ if for all $\alpha \geq 0$,*

$$\max_{X \sim X'} H_\alpha(\mathcal{M}(X)||\mathcal{M}(X')) \leq H_\alpha(P||Q).$$

Using dominating pairs of distributions, we can obtain $\delta(\varepsilon)$-upper bounds for adaptive compositions:

**Theorem 4** (Zhu et al. 2022). *If $(P, Q)$ dominates $\mathcal{M}$ and $(P', Q')$ dominates $\mathcal{M}'$, then $(P \times P', Q \times Q')$ dominates the adaptive composition $\mathcal{M} \circ \mathcal{M}'$.*

Having dominating pairs of distributions for each individual mechanism in a composition, the hockey-stick divergence can be transformed into a more easily computable form by using the privacy loss random variables (PRVs). PRV for a pair of distributions $(P, Q)$ is defined as follows.

**Definition 5.** *Let $P(t)$ and $Q(t)$ be probability density functions. We define the PRV $\omega_{P/Q}$ as*

$$\omega_{P/Q} = \log \frac{P(t)}{Q(t)}, \quad t \sim P(t),$$

*where $t \sim P(t)$ means that $t$ is distributed according to $P(t)$.*

With a slight abuse of notation, we denote the probability density function of the random variable $\omega_{P/Q}$ by $\omega_{P/Q}(t)$, and call it the privacy loss distribution (PLD).

The $\delta(\varepsilon)$-bounds can be stated using the following representation that involves the PRV.

**Theorem 6** (Gopi et al. 2021). *We have:*

$$H_{\mathrm{e}^\varepsilon}(P||Q) = \mathbb{E}_{\omega_{P/Q}} \left[ 1 - \mathrm{e}^{\varepsilon - \omega_{P/Q}} \right]_+, \tag{2.1}$$

*Moreover, if $\omega_{P/Q}$ is a PRV for the pair of distributions $(P, Q)$ and $\omega_{P'/Q'}$ a PRV for the pair of distributions $(P', Q')$, then the PRV for the pair of distributions $(P \times P', Q \times Q')$ is given by $\omega_{P/Q} + \omega_{P'/Q'}$*

By identifying dominating pairs of distributions for each mechanism in a composition and by formulating the $\delta(\varepsilon)$-bound via hockey-stick divergence as an integral of the form given in Equation 2.1, the numerical PLD accountants (Koskela et al., 2021; Gopi et al., 2021) can be utilised for computing accurate $\delta(\varepsilon)$-bounds.

We will also use the following subsampling amplification result (Proposition 30, Zhu et al., 2022), which leads to a privacy profile for the composed mechanism $\mathcal{M} \circ S_{Subset}$, where $S_{Subset}$ denotes a subsampling procedure where, from an input of $n$ entries, a fixed sized subset of $q \cdot n$, $0 < q \leq 1$, entries is sampled without replacement.

**Lemma 7** (Zhu et al. 2022). *Denote the subsampled mechanism $\widetilde{\mathcal{M}} := \mathcal{M} \circ S_{Subset}$. Suppose a pair of distributions $(P, Q)$ is a dominating pair of distributions for a mechanism $\mathcal{M}$ for all datasets of size $q \cdot n$ under the $\sim$-neighbouring relation (i.e., the substitute relation), where $q > 0$ is the subsampling ratio (size of the subset divided by $n$). Then, for all neighbouring datasets (under the $\sim$-neighbouring relation) $X$ and $Y$ of size $n$,*

$$H_\alpha\big(\widetilde{\mathcal{M}}(X)||\widetilde{\mathcal{M}}(Y)\big) \leq H_\alpha\big(q \cdot P + (1-q) \cdot Q||Q\big), \quad \text{for } \alpha \geq 1,$$
$$H_\alpha\big(\widetilde{\mathcal{M}}(X)||\widetilde{\mathcal{M}}(Y)\big) \leq H_\alpha\big(P||q \cdot Q + (1-q) \cdot P\big), \quad \text{for } 0 \leq \alpha < 1. \tag{2.2}$$

Considering the assumptions of Lemma 7, if we define a function $h : \mathbb{R}_{\geq 0} \to \mathbb{R}$

$$h(\alpha) = \max\{H_\alpha\big(q \cdot P + (1-q) \cdot Q||Q\big), H_\alpha\big(P||q \cdot Q + (1-q) \cdot P\big)\}, \tag{2.3}$$

we see that $h(\alpha)$ clearly defines a privacy profile: it is convex and has all the other required properties of a privacy profile. Thus we can use an existing numerical method (Doroshenko et al., 2022, Algorithm 1) with the function $h$ to obtain discrete-valued distributions $\widetilde{P}$ and $\widetilde{Q}$, that are a dominating pair for $\widetilde{\mathcal{M}} = \mathcal{M} \circ S_{Subset}$.

We remark that by (Theorem 10, Zhu et al., 2022), the two pairs of distributions on the right-hand side of Equation 2.2 give dominating pairs for remove and add neighbouring relations of datasets in case the pair $(P, Q)$ is a dominating pair of distributions for $\mathcal{M}$ under remove and add neighbouring relations, respectively, and can therefore be used to compute $(\varepsilon, \delta)$-upper bounds in case of add/remove neighbouring relations of datasets. Then, the computation is more straightforward since one can simply take the maximum of the $\delta(\varepsilon)$-values obtained under the remove and add neighbouring relations and therefore using the techniques of Doroshenko et al. (2022) is not necessary. We focus on using the $\sim$-relation as the dominating pair $(P, Q)$ obtained using both the post-processing results of (Feldman et al., 2023) and using our analysis for the $k$-RR local randomiser is a dominating pair under the $\sim$-relation. The $\sim$-relation is also behind the baseline bounds by Girgis et al. (2021). We illustrate in Fig. 1 the accuracy of the numerical construction of (Doroshenko et al., 2022, Algorithm 1) applied to the privacy profile given in Equation 2.3.

## 2.2 Numerical PLD accounting using FFT

In order to evaluate integrals of the form given in Equation 2.1, we use the Fast Fourier Transform (FFT)-based method by Koskela et al. (2021) called the Fourier Accountant (FA). This means that each PLD is truncated and placed on an equidistant numerical grid over an interval $[-L, L]$, $L > 0$. The distributions for the sums of the PRVs are given by convolutions of the individual PLDs and are evaluated using the FFT algorithm. By a careful error analysis the error incurred by the numerical method can be bounded and an upper $\delta(\varepsilon)$-bound obtained. We note that alternatively, for accurately computing the integrals we could also use the FFT-based method proposed by Gopi et al. (2021).

## 3 General shuffled $\varepsilon_0$-LDP mechanisms

Feldman et al. (2023) consider general $\varepsilon_0$-LDP local randomisers combined with a shuffler. The analysis allows also sequential adaptive compositions of the user contributions before shuffling. The analysis is based on decomposing individual LDP contributions to mixtures of data dependent part and noise, which leads to finding $(\varepsilon, \delta)$-bounds for the pair of 2-dimensional random variables (see Thm. 3.1 of Feldman et al., 2023)

$$P = (A + \Delta_1, C - A + \Delta_2), \qquad Q = (A + \Delta_2, C - A + \Delta_1), \tag{3.1}$$

where for $n \in \mathbb{N}$,

$$C \sim \mathrm{Bin}(n-1, 2p), \quad A \sim \mathrm{Bin}\left(C, \tfrac{1}{2}\right), \quad \Delta_1 \sim \mathrm{Bern}\left(\mathrm{e}^{\varepsilon_0} p\right) \quad \text{and} \quad \Delta_2 \sim \mathrm{Bin}\left(1 - \Delta_1, \tfrac{p}{1 - \mathrm{e}^{\varepsilon_0} p}\right), \tag{3.2}$$

and $p = \frac{1}{\mathrm{e}^{\varepsilon_0} + 1}$. Intuitively, $C$ denotes the number of other users whose mechanism outputs are indistinguishable "clones" of the two differing users, with $A$ denoting random split between these. Using the following lemma, we can use the FFT-based numerical accountants to obtain accurate bounds also for adaptive compositions of general $\varepsilon_0$-LDP shuffling mechanisms:

**Lemma 8.** *Let $X$ and $X'$ be neighbouring datasets and denote by $\mathcal{A}_s(X)$ and $\mathcal{A}_s(X')$ outputs of the shufflers of adaptive $\varepsilon_0$-LDP local randomisers (for a detailed description of $\mathcal{A}_s$, see Thm. 3.1 by Feldman et al., 2023, which uses the same notation). Then, for all $\alpha \geq 0$,*

$$H_\alpha(\mathcal{A}_s(X)||\mathcal{A}_s(X')) \leq H_\alpha(P||Q),$$

*where $P$ and $Q$ are given in Equation 3.1.*

*Proof.* By Thm. 3.1 of Feldman et al. (2023) there exists a post-processing algorithm $\Phi$ such that $\mathcal{A}_s(X)$ is distributed identically to $\Phi(P)$ and $\mathcal{A}_s(X')$ identically to $\Phi(Q)$. The claim follows then from the data-processing inequality which holds for the hockey-stick divergence (Balle et al., 2020a). $\qquad\square$

**Corollary 9.** *The pair of distributions $(P, Q)$ given in Equation 3.1 is a dominating pair of distributions for the shuffling mechanism $\mathcal{A}_s(X)$.*

Furthermore, using Thm. 4, we can bound the $\delta(\varepsilon)$ of $n_c$-wise adaptive composition of the shuffler $\mathcal{A}_s$ using product distributions of $P$s and $Q$s:

**Corollary 10.** *Denote $\mathcal{A}_s^{n_c}(X, z_0) = \mathcal{A}_s(X, \mathcal{A}_s(X, ...\mathcal{A}_s(X, z_0)))$ for some initial state $z_0$. For all neighbouring datasets $X$ and $X'$ and for all $\alpha \geq 0$,*

$$H_\alpha(\mathcal{A}_s^{n_c}(X) \| \mathcal{A}_s^{n_c}(X')) \leq H_\alpha(P \times \ldots \times P \| Q \times \ldots \times Q), \tag{3.3}$$

We remark that the case of heterogeneous adaptive compositions (e.g. varying $n$ and $\varepsilon_0$) can be handled analogously using Thm. 4.

Thus, using the bound of Equation 3.3 for $\alpha = e^\varepsilon$, we get upper bounds for adaptive compositions of general shuffled $\varepsilon_0$-LDP mechanisms with the Fourier accountant by finding the PLD for the distributions $P, Q$ (given in Equation 3.1). Note that even though the resulting $(\varepsilon, \delta)$-bound is tight for $P$'s and $Q$'s, it need not be tight for a specific mechanism like the shuffled $k$-RR. The bound simply gives an upper bound for any shuffled $\varepsilon_0$-LDP mechanisms.

### 3.1 PLD for shuffled $\varepsilon_0$-LDP mechanisms

To analyse compositions of general shuffled $\varepsilon_0$-LDP mechanisms, we need to form the PLD $\omega_{P/Q}$ determined by $P$ and $Q$ of Equation 3.1. Denoting $q = e^{\varepsilon_0} p$ and $\widetilde{q} = \frac{p}{1 - e^{\varepsilon_0} p}$, $p = \frac{1}{e^{\varepsilon_0} + 1}$, and writing out the randomness of $\Delta_1$ and $\Delta_2$ as mixtures, we see that the random variables $P$ and $Q$ given in Equation 3.1 can be expressed as

$$P = q \cdot P_0 + (1 - q)\widetilde{q} \cdot P_1 + (1 - q)(1 - \widetilde{q}) \cdot P_2, \quad Q = (1 - q)\widetilde{q} \cdot P_0 + q \cdot P_1 + (1 - q)(1 - \widetilde{q}) \cdot P_2,$$

where

$$P_0 \sim (A + 1, C - A), \quad P_1 \sim (A, C - A + 1), \quad P_2 = (A, C - A)$$

and $A$ and $C$ are as given in Equation 3.2. In the Appendix, we give the required expressions to determine the discrete-valued PLD

$$\omega_{P/Q}(s) = \sum_{a,b} \mathbb{P}(P = (a, b)) \cdot \delta_{s_{(a,b)}}(s), \quad s_{(a,b)} = \log\left(\frac{\mathbb{P}(P = (a,b))}{\mathbb{P}(Q = (a,b))}\right), \tag{3.4}$$

where $\delta_s(\cdot)$, $s \in \mathbb{R}$, denotes the Dirac delta function centred at $s$, and similarly also for $\omega_{Q/P}(s)$.

### 3.2 Lowering PLD computational complexity using Hoeffding's inequality

The PLD of Equation 3.4 has $\mathcal{O}(n^2)$ terms, which makes its naive evaluation overly expensive for a large number of users $n$. Using an appropriate tail bound (Hoeffding) for the binomial distribution, we can truncate part of the probability mass and add it directly to $\delta$. More specifically, if each PLD $\omega_i$, $1 \leq i \leq n_c$, in an $n_c$-wise composition is approximated by a truncated distribution $\widetilde{\omega}_i$ such that the truncated probability masses are $\tau_i \geq 0$, respectively, then $\delta(\varepsilon) = \widetilde{\delta}(\varepsilon) + \delta(\infty)$, where $\widetilde{\delta}(\varepsilon)$ is the value of the integral in Equation 2.1 obtained with the truncated PLDs and $\delta(\infty) = 1 - \prod_i(1 - \tau_i) \leq \sum_i \tau_i$, gives an upper bound for the composition without truncations. Using the Hoeffding's inequality we obtain an accurate approximation of $\omega_{P/Q}$ with only $\mathcal{O}(n)$ terms. We formalise this approximation as follows:

**Lemma 11.** *Let the PLD $\omega_{P/Q}$ be defined as in Equation 3.4 (Equation 3.1 gives $P$ and $Q$ which include $C \sim \text{Bin}(n - 1, 2p)$ and $A \sim \text{Bin}\left(C, \frac{1}{2}\right)$ ) and let $\tau > 0$. Consider the set*

$$S_n = [\max\left(0, (2p - c_n)(n - 1)\right), \min\left(n - 1, (2p + c_n)(n - 1)\right)],$$

*where $c_n = \sqrt{\frac{\log(4/\tau)}{2(n-1)}}$ and the set*

$$S_i = [\max\left(0, (\tfrac{1}{2} - c_i) \cdot i\right), \min\left(n - 1, (\tfrac{1}{2} + c_i) \cdot i\right)],$$

*where $c_i = \sqrt{\frac{\log(4/\tau)}{2 \cdot i}}$. Then, $\widetilde{\omega}_{P/Q}$ defined by*

$$\widetilde{\omega}_{P/Q}(s) = \sum_{i \in S_n} \sum_{j \in S_i} \mathbb{P}(P = (j+1, i-j)) \cdot \delta_{s_{j+1,i-j}}(s), \quad s_{a,b} = \log\left(\frac{\mathbb{P}(P=(a,b))}{\mathbb{P}(Q=(a,b))}\right) \tag{3.5}$$

*has $\mathcal{O}(n \cdot \log(4/\tau))$ terms and differs from $\omega_{P/Q}$ at most by mass $\tau$.*

*Proof.* As $A$ is conditioned on $C$, we first use a tail bound on $C$ and then on $A$ to reduce the number of terms. Using Hoeffding's inequality for $C \sim \text{Bin}(n-1, 2p)$ states that for $c > 0$,

$$\mathbb{P}\big(C \le (2p - c)(n-1)\big) \le \exp\big(-2(n-1)c^2\big),$$
$$\mathbb{P}\big(C \ge (2p + c)(n-1)\big) \le \exp\big(-2(n-1)c^2\big).$$

Requiring that $2 \cdot \exp\left(-2(n-1)c^2\right) \le \tau/2$ gives the condition $c \ge \sqrt{\frac{\log(4/\tau)}{2(n-1)}}$ and the expressions for $c_n$ and $S_n$. Similarly, we use Hoeffding's inequality for $A \sim \text{Bin}(C, \frac{1}{2})$ and get expressions for $c_i$ and $S_i$. The total neglected mass is at most $\tau/2 + \tau/2 = \tau$. For the number of terms, we see that $S_n$ contains at most $2c_n(n-1) = \sqrt{n-1}\sqrt{2 \cdot \log(4/\tau)}$ terms and for each $i$, and $S_i$ contains at most $2c_i i = \sqrt{i}\sqrt{2 \cdot \log(4/\tau)} \le \sqrt{n-1}\sqrt{2 \cdot \log(4/\tau)}$ terms. Thus $\widetilde{\omega}_{P/Q}$ has at most $\mathcal{O}(n \cdot \log(4/\tau))$ terms. We get the form of Equation 3.5 by an appropriate change of variables. $\square$

When evaluating $\widetilde{\omega}_{P/Q}$, we require that the neglected mass is smaller than some prescribed tolerance $\tau$ (e.g. $\tau = 10^{-12}$). When computing guarantees for compositions, the cost of FFT for evaluating the convolutions dominates the rest of the computation.

### 3.3 Experimental comparison to RDP

Figure 1 shows a comparison between the PLD and RDP applied to the pair of distributions $P$ and $Q$ given in Equation 3.1. RDP bounds for composition are computed using standard composition results (Mironov, 2012) and the RDP bounds are converted to DP bounds using the conversion formula given by Canonne et al. (2020). Naive evaluation of RDP-values is $\mathcal{O}(n^2)$ computation. We heuristically speed up RDP evaluation using the Hoeffding inequality (Lemma 11) and check that increasing the accuracy does not change the results.

### 3.4 Experimental comparison to the subsampled RDP bounds of Girgis et al. (2021)

Girgis et al. (2021) consider a protocol where a randomly sampled, fixed sized subset of users sends contributions to the shuffler on each round, and the local randomisers are assumed to be integer-valued $\varepsilon_0$-LDP mechanisms. This can be seen as a composition of a shuffler and a subsampling mechanism. We can generalise our analysis to this case via Lemma 7, and use Algorithm 1 of Doroshenko et al. (2022) on the function $h(\alpha)$ defined in Equation 2.3 to obtain the dominating pair of distributions for the subsampled shuffler. To this end, we need to define a grid for $\alpha$: $\{\alpha_0, \ldots, \alpha_{n_\alpha+1}\}$, where $0 = \alpha_0 < \alpha_1 < \ldots < \alpha_{n_\alpha} < \alpha_{n_\alpha+1} = \infty$. We consider a logarithmically equidistant grid between $\alpha_1$ and $\alpha_{n_\alpha}$. Thus, in practice this means that we need to determine $\alpha_1$ and $\alpha_{n_\alpha}$ and the value $n_\alpha$. Figure 2 illustrates the convergence of the obtained approximation as we refine the $\alpha$-grid, for a subsampled shuffler, where the dominating pair of distributions $P$ and $Q$ for the non-subsampled shuffler are obtained from (Thm. 3.1 Feldman et al., 2023).

As we see from Figure 3, this approach leads to considerably lower $\varepsilon(\delta)$-bounds than the approach by Girgis et al. (2021). Notice that the tightness of the PLD-based bound is mostly determined by the analysis of Feldman et al. (2023) which gives the dominating pair $(P, Q)$ of Equation 3.1 and that the RDP-based analysis of Girgis et al. (2021) is fundamentally different.

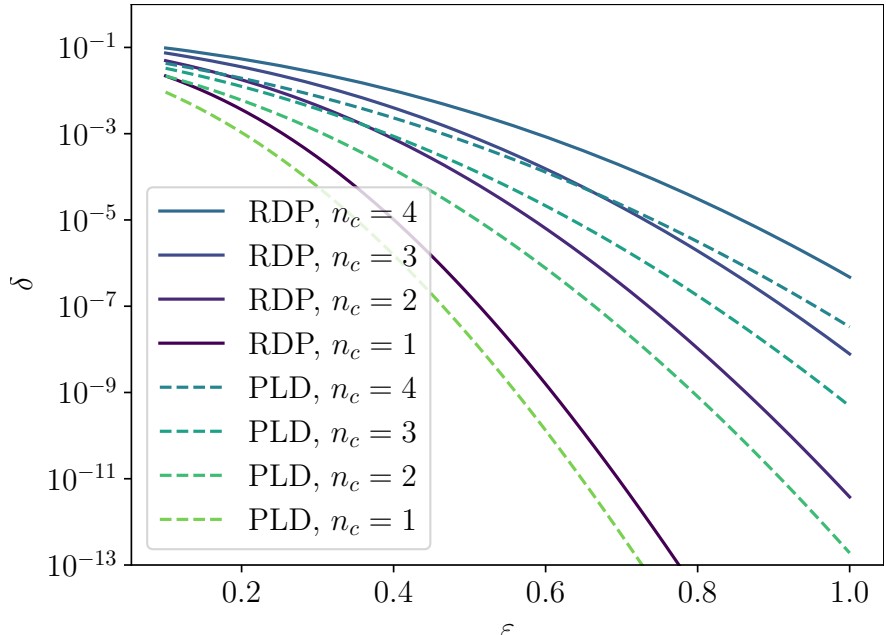

Figure 1: Evaluation of $\delta(\varepsilon)$ for general single and composite shuffle $\varepsilon_0$-LDP mechanisms using RDP accounting and FFT-based numerical accounting (PLD) applied to the pair of distributions $P$ and $Q$ given by the post-processing result of Feldman et al. (2023). Number of users $n = 10^4$ and the LDP parameter $\varepsilon_0 = 4.0$.

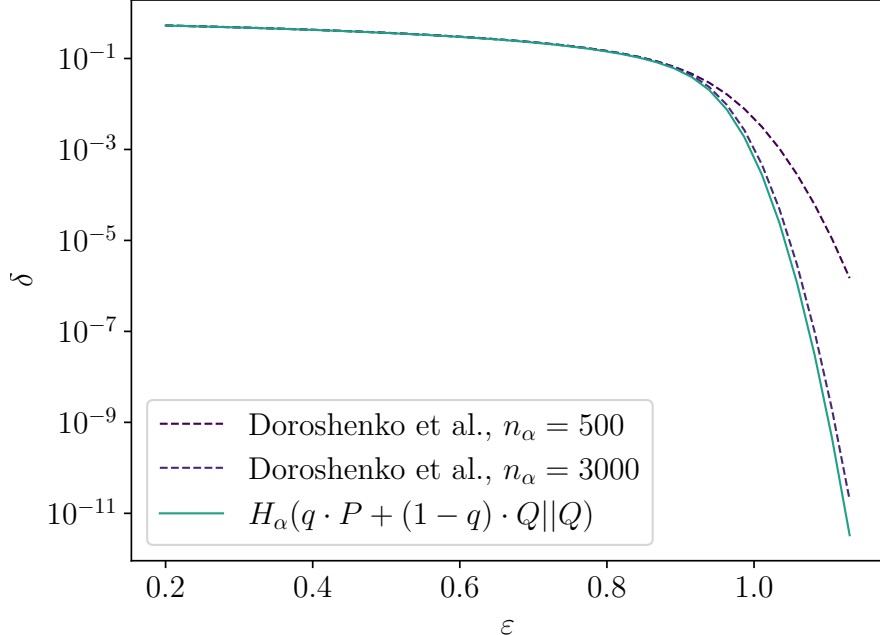

Figure 2: We apply FFT-based method on the dominating pair of distributions given by Algorithm 1 of Doroshenko et al. (2022) applied on the function $h(\alpha)$ that we obtain from Lemma 7, for different sizes of $\alpha$-grids. Here, the underlying $P$ and $Q$ are obtained from the analysis of Feldman et al. (2023), and we set $\varepsilon_0 = 3.0$, $n = 10^4$, $n_c = 2000$, subsampling ratio $q = 0.01$, $\alpha_1 = \exp(-0.25)$, $\alpha_{n_\alpha} = \exp(0.25)$, and take a logarithmically equidistant $\alpha$-grid. We also plot $H_{e^\varepsilon}(q \cdot P + (1-q) \cdot Q || Q)$ for comparison.

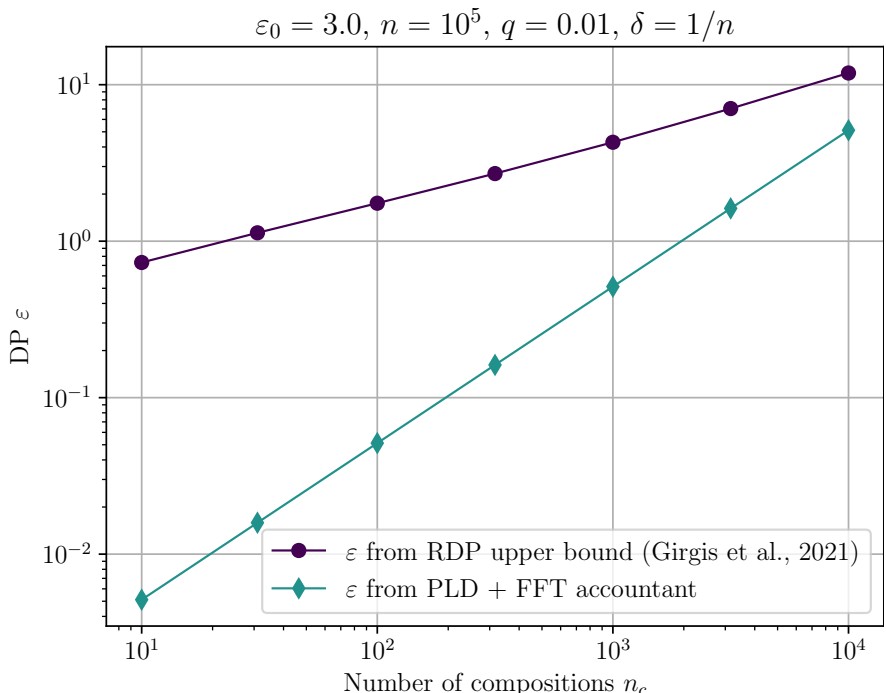

Figure 3: Evaluation of $\varepsilon(\delta)$ for compositions of subsampled shufflers of $\varepsilon_0$-local randomisers. We compare the bounds obtained using the FFT-accounting and the PLD determined by the numerical method of (Doroshenko et al., 2022, Algorithm 1) applied to the dominating pair of Equation 3.1 and the RDP-bounds given in Thm. 2 of (Girgis et al., 2021) that are mapped to $\varepsilon(\delta)$-bounds using Lemma 1 of (Girgis et al., 2021). Here the number of compositions $n_c$ varies and $n$ is fixed. Here $q$ denotes the subsampling ratio.

## 4 Shuffled $k$-randomised response

Balle et al. (2019) give a protocol for $n$ parties to compute a private histogram over the domain $[k]$ in the single-message shuffle model. The randomiser is parameterised by a probability $\gamma$, and consists of a $k$-ary randomised response mechanism ($k$-RR) that returns the true value with probability $1 - \gamma$ and a uniformly random value with probability $\gamma$. Denote this $k$-RR randomiser by $\mathcal{R}_{\gamma,k,n}^{PH}$ and the shuffling operation by $\mathcal{S}$. Thus, we are studying the privacy of the shuffled randomiser $\mathcal{M} = \mathcal{S} \circ \mathcal{R}_{\gamma,k,n}^{PH}$.

Consider first the proof of Balle et al. (2019, Thm. 3.1). Assuming without loss of generality that the differing data element between $X$ and $X'$, $X, X' \in [k]^n$, is $x_n$, the (strong) adversary $A_s$ used by Balle et al. (2019, Thm. 3.1) is defined as follows.

**Definition 12.** *Let $\mathcal{M} = \mathcal{S} \circ \mathcal{R}_{\gamma,k,n}^{PH}$ be the shuffled $k$-RR mechanism, and w.l.o.g. let the differing element be $x_n$. We define adversary $A_s$ as an adversary with the view*

$$View_{\mathcal{M}}^{A_s}(X) = \left( (x_1, \ldots, x_{n-1}), \quad \beta \in \{0,1\}^n, \quad (y_{\pi(1)}, \ldots, y_{\pi(n)}) \right),$$

*where $y$ are the outputs from the shuffler, $\beta$ is a binary vector identifying which parties answered randomly, and $\pi$ is a uniformly random permutation applied by the shuffler.*

Assuming w.l.o.g. that the differing element $x_n = 1$ and $x'_n = 2$, the proof then shows that for any possible view $V$ of the adversary $A_s$,

$$\frac{\mathbb{P}(View_{\mathcal{M}}^{A_s}(X) = V)}{\mathbb{P}(View_{\mathcal{M}}^{A_s}(X') = V)} = \frac{N_1 + 1}{N_2}, \tag{4.1}$$

where $N_i$ denotes the number of messages received by the server with value $i$ after removing from the output $Y$ any truthful answers submitted by the first $n - 1$ users. The $(\varepsilon, \delta)$-analysis of Balle et al. (2019) is based

on showing that for all neighbouring $X$ and $X'$,

$$\text{View}_{\mathcal{M}}^{A_s}(X) \simeq_{(\varepsilon,\delta)} \text{View}_{\mathcal{M}}^{A_s}(X') \tag{4.2}$$

for

$$\delta(\varepsilon) = \mathbb{P}\left(\frac{N_1 + 1}{N_2} \geq e^{\varepsilon}\right), \tag{4.3}$$

where the randomness of $(N_1, N_2)$ is determined by $\text{View}_{\mathcal{M}}^{A_s}(X)$. Instead of being mutually independent binomially distributed random variables as argued in the proof of Balle et al. (2019, Thm. 3.1), we claim that $N_1$ and $N_2$ are distributed as follows.

**Lemma 13.** *Let the* $\text{View}_{\mathcal{M}}^{A_s}(X)$ *be defined as in Def. 12. $N_1$ and $N_2$ denote the number of outcomes of the first $n-1$ local randomisers that are results of randomisation and equal 1 and 2, respectively. Then the counts $N_1$ and $N_2$ are distributed as*

$$(N_1, N_2) \sim (A, C),$$

*where $A \sim \text{Bin}(n-1, \frac{\gamma}{k})$ and $C \sim \text{Bin}(n-1-A, \frac{\gamma}{k-\gamma})$.*

*Proof.* First, more generally, consider $n-1$ independent trials and random variables for the numbers of observations for three classes: $N_1$, $N_2$ and a remainder class, with corresponding probabilities $p_1$, $p_2$ and $1 - p_1 - p_2$. Then, the multinomial probability gives

$$\mathbb{P}\big((N_1, N_2) = (n_1, n_2)\big)$$
$$= \frac{(n-1)!}{n_1! n_2! (n-1-n_1-n_2)!} p_1^{n_1} p_2^{n_2} (1 - p_1 - p_2)^{n-1-n_1-n_2}$$
$$= \frac{(n-1)!}{n_1!(n-1-n_1)!} p_1^{n_1}(1-p_1)^{n-1-n_1} \cdot \frac{(n-1-n_1)!}{n_2!(n-1-n_1-n_2)!} \frac{p_2^{n_2}(1-p_1-p_2)^{n-1-n_1-n_2}}{(1-p_1)^{n-1-n_1}}$$
$$= \frac{(n-1)!}{n_1!(n-1-n_1)!} p_1^{n_1}(1-p_1)^{n-1-n_1} \cdot \frac{(n-1-n_1)!}{n_2!(n-1-n_1-n_2)!} \left(\frac{p_2}{1-p_1}\right)^{n_2} \left(\frac{1-p_1-p_2}{1-p_1}\right)^{n-1-n_1-n_2}$$
$$= \frac{n!}{n_1!(n-1-n_1)!} p_1^{n_1}(1-p_1)^{n-1-n_1} \cdot \left[\frac{(n-1-n_1)!}{n_2!(n-1-n_1-n_2)!} \left(\frac{p_2}{1-p_1}\right)^{n_2} \left(1 - \frac{p_2}{1-p_1}\right)^{n-1-n_1-n_2}\right].$$

We recognise the probabilities of binomial distributions, and see that

$$(N_1, N_2) \sim (A, C),$$

where $A \sim \text{Bin}(n-1, p_1)$ and $C \sim \text{Bin}(n-1-A, \frac{p_2}{1-p_1})$. When $V \sim \text{View}_{\mathcal{M}}^{A_s}(X)$, we can think of $N_1$ and $N_2$ as numbers of outcomes of $n-1$ independent trials where both classes have probabilities $\gamma/k$. Substituting $p_1 = p_2 = \gamma/k$ in the above formula shows the claim.

$\square$

Using the reasoning of Balle et al. (2019, Thm. 3.1) (Equation 4.1), we can explicitly write the PLD which gives us tight $(\varepsilon, \delta)$-bounds. Recall from Def. 5 that the privacy loss random variable for $\text{View}_{\mathcal{M}}^{A_s}$ is given by

$$\omega_{X/X'}^{A_s} = \log\left(\frac{\mathbb{P}(\text{View}_{\mathcal{M}}^{A_s}(X) = V)}{\mathbb{P}(\text{View}_{\mathcal{M}}^{A_s}(X') = V)}\right), \quad V \sim \text{View}_{\mathcal{M}}^{A_s}(X). \tag{4.4}$$

Using this definition of PRV, Equation 4.1 and Lemma 13, we get the following.

**Theorem 14.** *Consider the adversary $A_s$ as given in Def 12. For all neighbouring datasets $X$ and $X'$, the PRV for $\text{View}_{\mathcal{M}}^{A_s}$ is given by*

$$\omega^{A_s} = \log\left(\frac{N_1 + 1}{N_2}\right),$$

*where*

$$(N_1, N_2) \sim (A, C),$$

*and $A \sim \text{Bin}(n-1, \frac{\gamma}{k})$ and $C \sim \text{Bin}(n-1-A, \frac{\gamma}{k-\gamma})$.*

Notice that this expression for $\omega^{A_s}$ is independent of any input to the local randomisers and holds for any neighbouring datasets $X$ and $X'$. Therefore it allows computing tight $\delta(\varepsilon)$-bounds for adaptive compositions of the $k$-RR shuffler in case we assume the adversary of Def. 12.

### 4.1 Tight bounds for weaker adversaries

Following the reasoning used for analysing the bounds against the adversary $A_s$ of Def. 12, we can compute tight $\delta(\varepsilon)$-bounds also for an adversary that has less information about the local randomisers. Having tight bounds also enables us to evaluate exactly how much different assumptions on the adversary cost us in terms of privacy. Instead of the adversary $A_s$ we analyse a weaker adversary $A_w$, who has extra information only on the first $n-1$ parties. We formalise this as follows.

**Definition 15.** *Let $\mathcal{M} = \mathcal{S} \circ \mathcal{R}_{\gamma,k,n}^{PH}$ be the shuffled $k$-RR mechanism, and w.l.o.g. let the differing element be $x_n$. Adversary $A_w$ is an adversary with the view*

$$View_{\mathcal{M}}^{A_w}(X) = \left((x_1, \ldots, x_{n-1}), \quad \beta \in \{0,1\}^{n-1}, \quad (y_{\pi(1)}, \ldots, y_{\pi(n)})\right),$$

*where $y$ are the outputs from the shuffler, $\beta$ is a binary vector identifying which of the first $n-1$ parties answered randomly, and $\pi$ is a uniformly random permutation applied by the shuffler.*

Note that compared to the stronger adversary $A_s$ formalised in Def. 12, the difference is only in the vector $\beta$. We write $b = \sum_i \beta_i$, and $B$ for the corresponding random variable in the following. The next theorem gives the random variables we need to calculate privacy bounds for adversary $A_w$.

**Theorem 16.** *Consider the adversary $A_w$ as given in Def 15. For all neighbouring datasets $X$ and $X'$, the PRV for $View_{\mathcal{M}}^{A_w}$ is given by*

$$\omega^{A_w} = \log\left(\frac{P_w}{Q_w}\right),$$

*where*

$$P_w = P_1 + P_2, \quad Q_w = Q_1 + Q_2, \tag{4.5}$$

*and*

$$P_1 \sim (1-\gamma) \cdot N_1 | B, \quad P_2 \sim \frac{\gamma}{k} \cdot (B+1),$$

$$Q_1 \sim (1-\gamma) \cdot N_2 | N_1, B, \quad Q_2 \sim \frac{\gamma}{k} \cdot (B+1),$$

$$B \sim \text{Bin}(n-1, \gamma),$$

$$N_1^B | B \sim \text{Bin}\left(B, \frac{1}{k}\right),$$

$$N_1 | B \sim N_1^B | B + \mathcal{R}_n,$$

$$\mathcal{R}_n \sim \text{Bern}(1 - \gamma + \gamma/k),$$

$$N_2 | N_1, B \sim \text{Bin}\left(B + 1 - N_1 | B, \frac{1}{k-1}\right).$$

As a direct corollary to this result, and analogously to the case of the adversary $A_s$, since the PLD $\omega^{A_w}$ is independent of any input to the local randomisers, we obtain tight $\delta(\varepsilon)$-bounds against the adversary $A_w$ for adaptive compositions using $\omega^{A_w}$.

### 4.2 Experimental comparison between specialized analysis of $k$-RR (Balle et al., 2019) and specialized Clones - analysis (Feldman et al., 2023)

In Figure 4 we compare the tight bounds obtained using the PRVs $\omega^{A_s}$ and $\omega^{A_w}$ with numerical FFT-based accounting, and the PLD obtained from the $k$-RR specific analysis of Feldman et al. (2023, Thm 5.2) combined with numerical accounting. We tune the parameters of the FFT-based numerical accounting so that the discretisation error is negligible. Notice that the underlying analysis of $k$-RR with the adversaries

$A_s, A_w$ has stronger assumptions about the adversary than the analysis by Feldman et al. (2023), as the adversaries know which of the messages were randomised (except for the differing element in case of the weaker adversary $A_w$). For the weaker adversary $A_w$, we already obtain stronger guarantees than by using the analysis of Feldman et al. (2023).

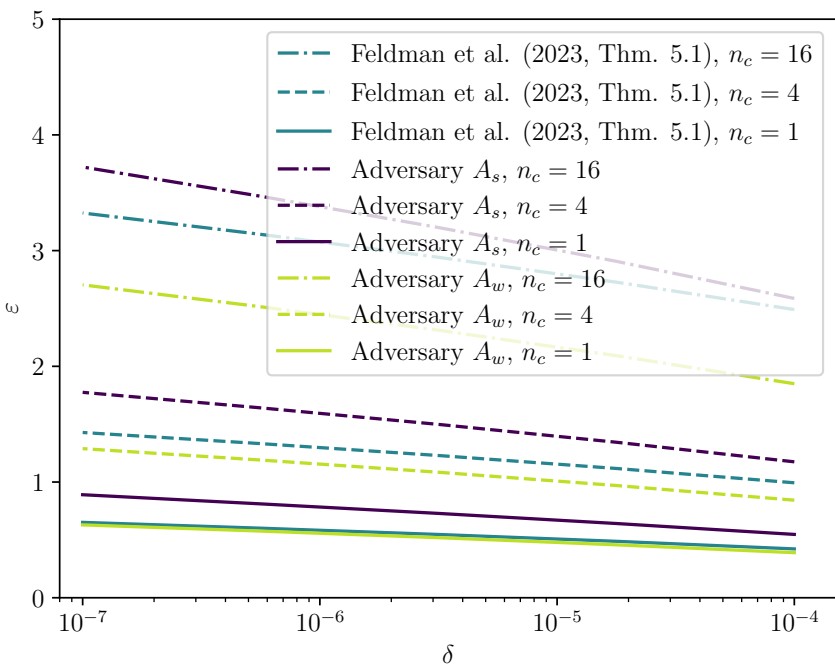

Figure 4: $k$-RR with the strong adversary $A_s$ (PRV $\omega^{A_s}$ determined by Thm 14) and the weak adversary (PRV $\omega^{A_s}$ determined by Thm 16) and tight $(\varepsilon, \delta)$-DP bounds obtained using FFT-accounting for different numbers of compositions $n_c$. Here $n = 1000$, probability of randomising $\gamma = 0.25$, and $k = 4$. Also shown are the bounds computed using the $k$-RR specific result by Feldman et al. (2023, Thm 5.2).

# 5   On the difficulty of obtaining bounds in the general case

We have provided means to compute accurate $(\varepsilon, \delta)$-bounds for the general $\varepsilon_0$-LDP shuffler using the results by Feldman et al. (2023) and tight bounds for the case of $k$-randomised response. Using the following example, we illustrate the computational difficulty of obtaining tight bounds for arbitrary local randomisers.

Consider neighbouring datasets $X, X' \in \mathbb{R}^n$, where all elements of $X$ are equal, and $X'$ contains one element differing by 1. Without loss of generality (due to shifting and scaling invariance of DP), we may consider the case where $X$ consists of zeros and $X'$ has 1 at some element. Considering a mechanism $\mathcal{M}$ that consists of adding Gaussian noise with variance $\sigma^2$ to each element and then shuffling, we see that the adversary sees the output of $\mathcal{M}(X)$ distributed as $\mathcal{M}(X) \sim \mathcal{N}(0, \sigma^2 I_n)$, and the output $\mathcal{M}(X')$ as the mixture distribution $\mathcal{M}(X') \sim \frac{1}{n} \cdot \mathcal{N}(e_1, \sigma^2 I_n) + \ldots + \frac{1}{n} \cdot \mathcal{N}(e_n, \sigma^2 I_n)$, where $e_i$ denotes the $i$th unit vector.

Determining the hockey-stick divergence $H_{e^\varepsilon}(\mathcal{M}(X')||\mathcal{M}(X))$ cannot be projected to a lower-dimensional problem, unlike in the case of the (subsampled) Gaussian mechanism, for example, which is equivalent to a one-dimensional problem. This means that in order to obtain tight $(\varepsilon, \delta)$-bounds, we need to numerically evaluate the $n$-dimensional hockey-stick integral $H_{e^\varepsilon}(\mathcal{M}(X')||\mathcal{M}(X))$.

Using a numerical grid as in FFT-based accountants is unthinkable due to the curse of the dimensionality. However, we may use the fact that for any dataset $X$, the density function $f_X(t)$ of $\mathcal{M}(X)$ is a permutation-invariant function, meaning that for any $t \in \mathbb{R}^n$ and for any permutation $\sigma \in \pi_n$, $f_X(\sigma(t)) = f_X(t)$. This allows reducing the number of required points on a regular grid for the hockey stick integral from $O(m^n)$ to $O(m^n/n!)$, where $m$ is the number of discretisation points in each dimension. Recent research on numerical

integration of permutation-invariant functions (e.g. Nuyens et al., 2016) suggests it may be possible to significantly reduce or even eliminate the dependence on $n$ using more advanced integration techniques.

In the Appendix C.2, we give results on experiments where we have computed $H_{e^\varepsilon}(\mathcal{M}(X')||\mathcal{M}(X))$ using Monte Carlo integration on a hypercube $[-L, L]^n$ which requires $\approx 5 \cdot 10^7$ samples for getting two correct significant figures already for $n = 7$.

## 6 Discussion

We have shown how numerical privacy accounting with privacy loss distributions can be used to calculate accurate upper bounds for the compositions of various $(\varepsilon, \delta)$-DP mechanisms, as well as for different adversaries in the shuffle model. An alternative accounting approach uses Rényi differential privacy (Mironov, 2017). We have carried out experimental comparisons between the RDP and the PLD approaches. As illustrated by the comparison against the results of Girgis et al. (2021) in Fig. 3, numerical PLD accounting can sometimes lead to considerably tighter bounds.

When comparing numerical and analytical privacy bounds, they are in many cases complementary and serve different purposes. Numerical accountants allow finding the tightest possible bounds for implementations and enable more unbiased comparison of algorithms when accuracy of accounting is not a factor. Analytical bounds enable theoretical research and understanding of scaling properties of algorithms, but the inaccuracy of the bounds raises the risk of misleading conclusions about privacy claims.

While our results provide improvements over previous state-of-the-art, they only provide optimal accounting for $k$-randomised response. Developing optimal accounting for more general mechanisms as well as extending the results to $(\varepsilon_0, \delta_0)$-LDP base mechanisms are important topics for future research.

## Acknowledgments

This work was supported by the Academy of Finland (Flagship programme: Finnish Center for Artificial Intelligence, FCAI; and grant 325573), the Strategic Research Council at the Academy of Finland (Grant 336032) as well as the European Union (Project 101070617). Views and opinions expressed are however those of the author(s) only and do not necessarily reflect those of the European Union or the European Commission. Neither the European Union nor the granting authority can be held responsible for them.

The authors would also like to thank the anonymous reviewers and the action editor at TMLR for noticing potential points of confusion in an earlier version, as well as for the helpful discussions.

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

# A   Auxiliary results for determining the PLD of general $\varepsilon_0$ shufflers

We recall the following from Section 3.1. Denoting $q = \mathrm{e}^{\varepsilon_0}p$ and $\widetilde{q} = \frac{p}{1 - \mathrm{e}^{\varepsilon_0}p}$, $p = \frac{1}{\mathrm{e}^{\varepsilon_0} + 1}$, the dominating pair of distributions $(P, Q)$ is determined are given by the mixtures

$$P = q \cdot P_0 + (1 - q)\widetilde{q} \cdot P_1 + (1 - q)(1 - \widetilde{q}) \cdot P_2, \quad Q = (1 - q)\widetilde{q} \cdot P_0 + q \cdot P_1 + (1 - q)(1 - \widetilde{q}) \cdot P_2,$$

where

$$P_0 \sim (A + 1, C - A), \quad P_1 \sim (A, C - A + 1), \quad P_2 = (A, C - A)$$

and for $n \in \mathbb{N}$, $A$ and $C$ are as

$$C \sim \mathrm{Bin}(n - 1, 2p), \quad A \sim \mathrm{Bin}\left(C, \tfrac{1}{2}\right), \quad \Delta_1 \sim \mathrm{Bern}\left(\mathrm{e}^{\varepsilon_0}p\right) \quad \text{and} \quad \Delta_2 \sim \mathrm{Bin}\left(1 - \Delta_1, \tfrac{p}{1 - \mathrm{e}^{\varepsilon_0}p}\right).$$

In this section we give the expressions needed to determine the PLD

$$\omega_{P/Q}(s) = \sum_{a,b} \mathbb{P}(P = (a, b)) \cdot \delta_{s_{a,b}}(s), \quad s_{a,b} = \log\left(\frac{\mathbb{P}(P = (a, b))}{\mathbb{P}(Q = (a, b))}\right), \tag{A.1}$$

and similarly also $\omega_{Q/P}$.

## A.1   Determining the log ratios $s_{a,b}$

To determine $s_{a,b}$'s, we need the following auxiliary results.

**Lemma A.1.** *When $b > 0$ and $a > 0$, we have:*

$$\mathbb{P}(P_0 = (a, b)) = \frac{a}{b} \cdot \mathbb{P}(P_1 = (a, b)).$$

*Proof.* We see that $P_0 = (a, b)$ if and only if $A = a - 1$ and $C = a + b - 1$. Since

$$\mathbb{P}(A = a - 1 \,|\, C = a + b - 1) = \binom{a + b - 1}{a - 1} \frac{1}{2^{a+b-1}}$$

$$= \frac{a}{b} \cdot \binom{a + b - 1}{a} \frac{1}{2^{a+b-1}}$$

$$= \frac{a}{b} \cdot \mathbb{P}(A = a \,|\, C = a + b - 1),$$

we see that

$$\mathbb{P}(P_0 = (a, b)) = \mathbb{P}(C = a + b - 1) \cdot \mathbb{P}(A = a - 1 \,|\, C = a + b - 1)$$

$$= \mathbb{P}(C = a + b - 1) \cdot \frac{a}{b} \cdot \mathbb{P}(A = a \,|\, C = a + b - 1)$$

$$= \frac{a}{b} \cdot \mathbb{P}(P_1 = (a, b)),$$

since $P_1 = (a, b)$ if and only if $A = a$ and $C = a + b - 1$. $\qquad\square$

**Lemma A.2.** *When $b > 0$ and $a > 0$, we have:*

$$\mathbb{P}(P_0 = (a, b)) = \frac{(1 - 2p)a}{(n - a - b)p} \cdot \mathbb{P}(P_2 = (a, b)).$$

*Proof.* We see that $P_2 = (a, b)$ if and only if $A = a$ and $C = a + b$. Since

$$\mathbb{P}(C = a + b) = \binom{n - 1}{a + b}(2p)^{a+b}(1 - 2p)^{n-1-a-b}$$

$$= \frac{2p}{1 - 2p}\binom{n - 1}{a + b}(2p)^{a+b-1}(1 - 2p)^{n-1-a-b+1}$$

$$= \frac{2p}{1 - 2p}\frac{n - a - b}{a + b}\binom{n - 1}{a + b - 1}(2p)^{a+b-1}(1 - 2p)^{n-1-a-b+1}$$

$$= \frac{2p}{1 - 2p}\frac{n - a - b}{a + b} \cdot \mathbb{P}(C = a + b - 1)$$

and since

$$\mathbb{P}(A = a \,|\, C = a + b - 1) = \binom{a+b-1}{a} \frac{1}{2^{a+b-1}} = \frac{2b}{a+b} \binom{a+b}{a} \frac{1}{2^{a+b}} = \frac{2b}{a+b} \cdot \mathbb{P}(A = a \,|\, C = a+b),$$

we see that

$$\begin{aligned}
\mathbb{P}(P_0 = (a,b)) &= \mathbb{P}(C = a+b-1) \cdot \mathbb{P}(A = a-1 \,|\, C = a+b-1) \\
&= \mathbb{P}(C = a+b-1) \cdot \frac{a}{b} \cdot \mathbb{P}(A = a \,|\, C = a+b-1) \\
&= \frac{(1-2p)a}{(n-a-b)p} \cdot \mathbb{P}(C = a+b) \cdot \frac{a}{b} \cdot \mathbb{P}(A = a \,|\, C = a+b) \\
&= \frac{(1-2p)a}{(n-a-b)p} \cdot \mathbb{P}(P_2 = (a,b)).
\end{aligned}$$

$\square$

As a corollary of Lemmas A.1 and A.2 we get the following expressions with which we can also determine the log ratios $s_{a,b}$.

**Corollary A.3.** *We have:*

$$\mathbb{P}\big(P = (a,b)\big) = \left[ q + (1-q)\widetilde{q} \cdot \frac{b}{a} + (1-q)(1-\widetilde{q}) \frac{(n-a-b)p}{(1-2p)a} \right] \cdot \mathbb{P}\big(P_0 = (a,b)\big)$$

*and*

$$\mathbb{P}\big(Q = (a,b)\big) = \left[ q \cdot \frac{b}{a} + (1-q)\widetilde{q} + (1-q)(1-\widetilde{q}) \frac{(n-a-b)p}{(1-2p)a} \right] \cdot \mathbb{P}\big(P_0 = (a,b)\big).$$

Probabilities for the cases $a = 0$ and $b = 0$ become extremely small already for moderate values of $n$. When using the Hoeffding inequality based $O(n)$-approximation to determine the PLDs, we do not need to evaluate these probabilities so we do not consider writing them out.

Corollary A.3 gives $\mathbb{P}\big(P = (a,b)\big)$ and $\mathbb{P}\big(Q = (a,b)\big)$ in terms of $\mathbb{P}(P_0 = (a,b))$, and that is given by the following expression which we get by change of variables.

**Lemma A.4.** *When $a > 0$,*

$$\mathbb{P}(P_0 = (a,b)) = \binom{n-1}{i} \binom{i}{j} p^i (1-p)^{n-1-i} \frac{1}{2^i},$$

*where $(a,b) = (j+1, i-j)$ (i.e., $C = i$ and $A = j$).*

## B  More detailed proof of the Lemma: Lowering PLD computational complexity using Hoeffding's inequality

Using an appropriate tail bound (Hoeffding) for the binomial distribution, we can truncate part of the probability mass and add it directly to $\delta$. More specifically, if each PLD $\omega_i$, $1 \leq i \leq n_c$, in an $n_c$-composition is approximated by a truncated distribution $\widetilde{\omega}_i$ such that the truncated probability masses are $\tau_i \geq 0$, respectively, then

$$\delta(\varepsilon) = \widetilde{\delta}(\varepsilon) + \delta(\infty),$$

where $\widetilde{\delta}(\varepsilon)$ is the value of the hockey-stick divergence obtained with the truncated PLDs $\widetilde{\omega}_i$, $1 \leq i \leq n_c$, and where

$$\delta(\infty) = 1 - \prod_i (1 - \tau_i) \leq \sum_i \tau_i,$$

gives an upper bound for the composition without truncations, see e.g. Thm 1 in Sommer et al. (2019). Using the Hoeffding inequality we obtain an accurate approximation of $\omega_{P/Q}$ (or $\omega_{Q/P}$) with only $\mathcal{O}(n)$ terms. We formalise this approximation as follows.

**Lemma 11.** *Let $\tau > 0$. Consider the set*

$$S_n = [\max\left(0, (2p - c_n)(n-1)\right), \min\left(n-1, (2p+c_n)(n-1)\right)],$$

*where $c_n = \sqrt{\frac{\log(4/\tau)}{2(n-1)}}$ and the set*

$$S_i = [\max\left(0, (\tfrac{1}{2} - c_i) \cdot i\right), \min\left(n-1, (\tfrac{1}{2} + c_i) \cdot i\right)],$$

*where $c_i = \sqrt{\frac{\log(4/\tau)}{2 \cdot i}}$. Then, the distribution $\widetilde{\omega}_{P/Q}$ defined by*

$$\widetilde{\omega}_{P/Q}(s) = \sum_{i \in S_n} \sum_{j \in S_i} \mathbb{P}(P = (j+1, i-j)) \cdot \delta_{s_{j+1,i-j}}(s), \quad s_{a,b} = \log\left(\frac{\mathbb{P}(P=(a,b))}{\mathbb{P}(Q=(a,b))}\right) \tag{B.1}$$

*has $\mathcal{O}(n \cdot \log(4/\tau))$ terms and differs from $\omega_{P/Q}$ at most mass $\tau$.*

*Proof.* Using Hoeffding's inequality for $C \sim \mathrm{Bin}(n-1, 2p)$ states that for $c > 0$,

$$\mathbb{P}\big(C \le (2p-c)(n-1)\big) \le \exp\big(-2(n-1)c^2\big),$$
$$\mathbb{P}\big(C \ge (2p+c)(n-1)\big) \le \exp\big(-2(n-1)c^2\big).$$

Requiring that $2 \cdot \exp\left(-2(n-1)c^2\right) \le \tau/2$ gives the condition $c \ge \sqrt{\frac{\log(4/\tau)}{2(n-1)}}$ and the expressions for $c_n$ and $S_n$. Similarly, we use Hoeffding's inequality for $A \sim \mathrm{Bin}(C, \frac{1}{2})$ and get expressions for $c_i$ and $S_i$. The total neglegted mass is at most $\tau/2 + \tau/2 = \tau$. For the number of terms, we see that $S_n$ contains at most $2c_n(n-1) = \sqrt{n-1}\sqrt{2 \cdot \log(4/\tau)}$ terms and for each $i$, $S_i$ contains at most $2c_i i = \sqrt{i}\sqrt{2 \cdot \log(4/\tau)} \le \sqrt{n-1}\sqrt{2 \cdot \log(4/\tau)}$ terms. Thus $\widetilde{\omega}_{P/Q}$ has at most $\mathcal{O}(n \cdot \log(4/\tau))$ terms. We get the expression of Equation B.1 by the change of variables $a = i + 1$ $(A = i)$ and $b = i - j$ $(C = j)$. $\qquad\square$

## C Auxiliary results for Section 4

### C.1 Proof of Theorem 16

**Theorem C.1.** *Consider the adversary $A_w$ as given in Def 15. For all neighbouring datasets $X$ and $X'$, the PRV for $\mathrm{View}_{\mathcal{M}}^{A_w}$ is given by*

$$\omega^{A_w} = \log\left(\frac{P_w}{Q_w}\right),$$

*where*

$$P_w = P_1 + P_2, \quad Q_w = Q_1 + Q_2, \tag{C.1}$$

*and*

$$P_1 \sim (1 - \gamma) \cdot N_1 | B, \quad P_2 \sim \frac{\gamma}{k} \cdot (B + 1),$$

$$Q_1 \sim (1 - \gamma) \cdot N_2 | N_1, B, \quad Q_2 \sim \frac{\gamma}{k} \cdot (B + 1),$$

$$B \sim \mathrm{Bin}(n - 1, \gamma),$$

$$N_1^B | B \sim \mathrm{Bin}\left(B, \frac{1}{k}\right),$$

$$N_1 | B \sim N_1^B | B + \mathcal{R}_n,$$

$$\mathcal{R}_n \sim \mathrm{Bern}(1 - \gamma + \gamma/k),$$

$$N_2 | N_1, B \sim \mathrm{Bin}\left(B + 1 - N_1 | B, \frac{1}{k-1}\right).$$

*Proof.* Assume w.l.o.g. that the differing elements are $x_n = 1, x'_n = 2$. Notice that for $k$-RR, seeing the shuffler output is equivalent to seeing the total counts for each class resulting from applying the local randomisers to $X$ or $X'$. The adversary $A_w$ can remove all truthfully reported values by client $j$, $j \in [n-1]$. Denote the observed counts after this removal by $n_i, i = 1, \ldots, k$, so $\sum_{i=1}^{k} n_i = b + 1$.

We now have

$$\mathbb{P}(\text{View}_{\mathcal{M}}^{A_w}(\mathbf{x}) = V) = \sum_{i=1}^{k} \mathbb{P}(N_1 = n_1, \ldots, N_i = n_i - 1, N_{i+1} = n_{i+1}, \ldots N_k = n_k | B) \cdot \mathbb{P}(\mathcal{R}(x_n) = i) \cdot \mathbb{P}(B = b)$$

$$= \binom{b}{n_1 - 1, n_2, \ldots, n_k} \left(\frac{1}{k}\right)^b \cdot \left(1 - \gamma + \frac{\gamma}{k}\right) \cdot \gamma^b (1-\gamma)^{n-1-b}$$

$$+ \sum_{i=2}^{k} \binom{b}{n_1, \ldots, n_i - 1, n_{i+1}, \ldots, n_k} \left(\frac{1}{k}\right)^b \cdot \frac{\gamma}{k} \cdot \gamma^b (1-\gamma)^{n-1-b}$$

$$= \binom{b}{n_1, n_2, \ldots, n_k} \frac{\gamma^b (1-\gamma)^{n-1-b}}{k^b} \left[n_1(1 - \gamma + \frac{\gamma}{k}) + \sum_{i=2}^{k} n_i \frac{\gamma}{k}\right]$$

$$= \binom{b}{n_1, n_2, \ldots, n_k} \frac{\gamma^b (1-\gamma)^{n-1-b}}{k^b} \left[n_1(1 - \gamma + \frac{\gamma}{k}) + (b + 1 - n_1)\frac{\gamma}{k}\right]$$

$$= \binom{b}{n_1, n_2, \ldots, n_k} \frac{\gamma^b (1-\gamma)^{n-1-b}}{k^b} \left[n_1(1 - \gamma) + \frac{\gamma}{k}(b + 1)\right].$$

(C.2)

Noting then that $\mathbb{P}(\mathcal{R}_{\gamma,k,n}^{PH}(x'_n) = i) = (1 - \gamma + \frac{\gamma}{k})$ when $i = 2$ and $\frac{\gamma}{k}$ otherwise, repeating essentially the same steps gives

$$\mathbb{P}(\text{View}_{\mathcal{M}}^{A_w}(X') = V) = \binom{b}{n_1, n_2, \ldots, n_k} \frac{\gamma^b (1-\gamma)^{n-1-b}}{k^b} \left[n_2(1 - \gamma) + \frac{\gamma}{k}(b + 1)\right]. \qquad (C.3)$$

Looking at the ratio of the two final probabilities given in Equation C.2 and in Equation C.3, we get

$$\frac{\mathbb{P}(\text{View}_{\mathcal{M}}^{A_w}(X) = V)}{\mathbb{P}(\text{View}_{\mathcal{M}}^{A_w}(X') = V)} = \frac{N_1 | B \cdot (1 - \gamma) + \frac{\gamma}{k}(B + 1)}{N_2 | N_1, B \cdot (1 - \gamma) + \frac{\gamma}{k}(B + 1)},$$

where we write, e.g., $N_1 | B$ for the random variables $N_1$ conditional on $B$. This shows that for DP bounds, the adversaries' full view is equivalent to only considering the joint distribution of $(N_1, N_2, B)$, and we can therefore look at the neighbouring random variables

$$P_w = P_1 + P_2, \quad Q_w = Q_1 + Q_2, \qquad (C.4)$$

where

$$P_1 \sim (1 - \gamma) \cdot N_1 | B, \quad P_2 \sim \frac{\gamma}{k} \cdot (B + 1),$$

$$Q_1 \sim (1 - \gamma) \cdot N_2 | N_1, B, \quad Q_2 \sim \frac{\gamma}{k} \cdot (B + 1).$$

Writing $N_i^B$, $i = 1, 2$, for the count in class $i$ resulting from the noise sent by the $n - 1$ parties, and denoting by $\mathcal{R}_n$ a Bernoulli random variable s.t. $\mathcal{R}_n = 1$, if $\mathcal{R}(x_n) = 1$, similarly to the proof in case of the strong adversary, we have

$$B \sim \text{Bin}(n - 1, \gamma), \qquad N_1^B | B \sim \text{Bin}\left(B, \frac{1}{k}\right), \qquad \mathcal{R}_n \sim \text{Bern}(1 - \gamma + \gamma/k). \qquad (C.5)$$

As $V \sim \text{View}_{\mathcal{M}}^{A_w}(X)$, and

$$N_2 | N_1, \mathcal{R}_n, B \sim \begin{cases} \text{Bin}(B + 1 - N_1 | B, \frac{1}{k-1}), & \text{if } \mathcal{R}_n = 1, \\ \text{Bin}(B - N_1 | B, \frac{1}{k-1}) + \text{Bern}(\frac{1}{k-1}), & \text{if } \mathcal{R}_n = 0, \end{cases}$$

we finally have

$$N_1|B = N_1^B|B + \mathcal{R}_n, \quad N_2|N_1, B = \text{Bin}(B + 1 - N_1|B, \frac{1}{k-1}). \tag{C.6}$$

The distributions of Equation C.5 and Equation C.6 determine the neighbouring distributions $P_w$ and $Q_w$ that are given in Equation C.4. This completes the proof. □

### C.2 Experiment for Section 5

Consider neighbouring datasets $X, X' \in \mathbb{R}^n$, where all elements of $X$ are equal, and $X'$ contains one element differing by 1. Without loss of generality (due to shifting and scaling invariance of DP), we may consider the case where $X$ consists of zeros and $X'$ has 1 at some element. Considering a mechanism $\mathcal{M}$ that consists of adding Gaussian noise with variance $\sigma^2$ to each element and then shuffling, we see that the adversary sees the output of $\mathcal{M}(X)$ distributed as $\mathcal{M}(X) \sim \mathcal{N}(0, \sigma^2 I_n)$, and the output $\mathcal{M}(X')$ as the mixture distribution $\mathcal{M}(X') \sim \frac{1}{n} \cdot \mathcal{N}(e_1, \sigma^2 I_n) + \ldots + \frac{1}{n} \cdot \mathcal{N}(e_n, \sigma^2 I_n)$, where $e_i$ denotes the $i$th unit vector. In order to obtain tight $(\varepsilon, \delta)$-bounds, we need to numerically evaluate the $n$-dimensional hockey-stick integral $H_{e^\varepsilon}(\mathcal{M}(X')||\mathcal{M}(X))$.

In Figure 5 we have computed $H_{e^\varepsilon}(\mathcal{M}(X')||\mathcal{M}(X))$ up to $n = 7$ using Monte Carlo integration on a hypercube $[-L, L]^n$ which requires $\approx 5 \cdot 10^7$ samples for getting two correct significant figures for $n = 7$.

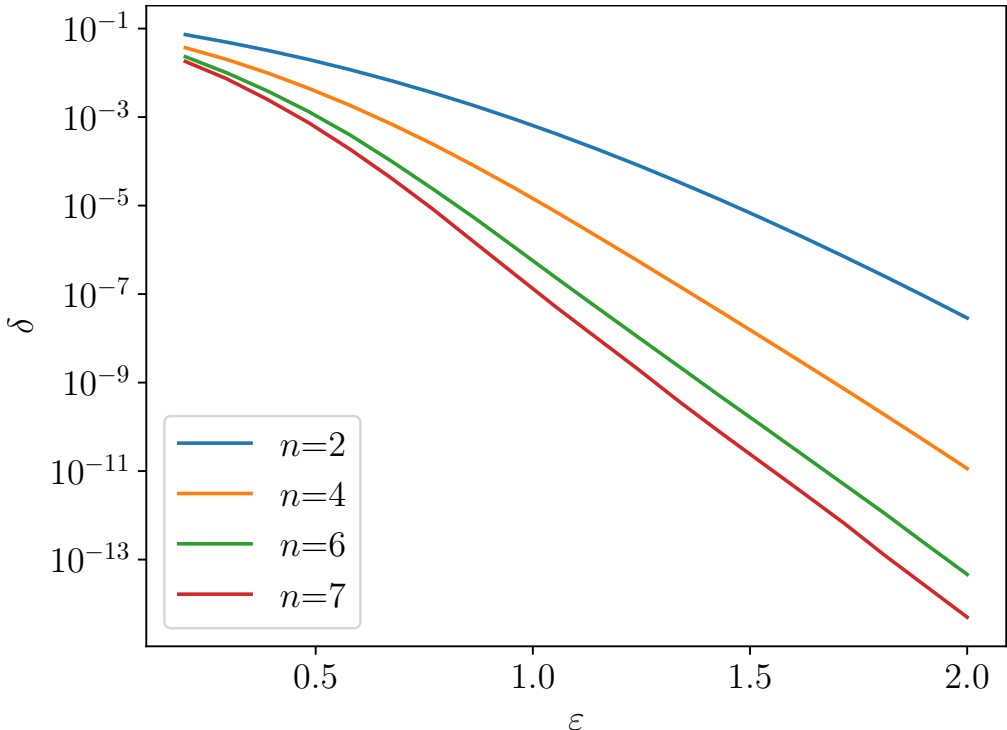

Figure 5: Approximation of tight $\delta(\varepsilon)$ for shuffled outputs of Gaussian mechanisms ($\sigma = 2.0$) by Monte Carlo integration of the hockey-stick divergence $H_{e^\varepsilon}(\mathcal{M}(X')||\mathcal{M}(X))$, using $5 \cdot 10^7$ samples (two correct significant figures).

