# OpenReview forum: "Numerical Accounting in the Shuffle Model of Differential Privacy"
_TMLR — Accepted by TMLR_

### Review · Reviewer_KLwm · 2022-11-04

**Summary Of Contributions:**

By relying on recent theoretical characterizations, the paper adapts existing numerical methods for tight(er) privacy loss accounting to LDP shufflers and shufflers for k-randomized response mechanisms. It is shown in simulation that the proposed method is superior to existing analytical and numerical baselines.

**Audience:**

Yes

**Broader Impact Concerns:**

No concerns.

**Claims And Evidence:**

Yes

**Requested Changes:**

The paper already looks good overall, please address the stylistic comments if they make sense to you. I have made several other suggestions that I think could improve the paper but they are not critical.

**Strengths And Weaknesses:**

This is a solid contribution, the results are convincing and the paper is easy to read. I believe the paper has great potential for practical deployments of DP. To the best of my knowledge all relevant baselines are included in the comparisons and the proposed accountant is indeed superior to existing work. I have some questions and suggestions:
1. In my opinion, the title of the paper can be made more indicative of the content. In particular, an important aspect of the paper is the fact that the accounting is numerical rather than analytical. I would probably say "tight numerical accounting ..." or something like that. If the other reviewers and the authors don't think this would help situate the paper better, then please ignore the suggestion. On a related note, I didn't love seeing the word "tight" in the title because the accountant seems to be tight only in the k-randomized response setting; is this correct? (For example, the Hoeffding truncation is not tight.) But I can see why the word tight is justified since the analysis is almost tight.
2. I believe you could make your analysis even tighter by replacing the Hoeffding confidence intervals in Lemma 11 by tighter confidence intervals for the binomial. In particular, you could just numerically invert an exact test for the binomial distribution or, say, use the Clopper–Pearson interval. This approach wouldn't lead to a nice analytical expression like Hoeffding's inequality but Hoeffding confidence intervals are generally quite loose and your method is numerical anyway, so you may be able to improve. I am not sure if this would lead to a visible improvement or not, but it may be worth a shot.
3. What is p in Lemma 11? I don't remember seeing the definition.
4. Minor: you refer to omega-til as a distribution (for example, in equation (3.5) in Lemma 11) but it's not actually a distribution since you don't renormalize it to add up to 1.
5. Some stylistic comments:
- In Theorem 6, why wouldn't you just write e^{eps - omega_{P/Q}}? Since you define omega_{P/Q} as a random variable anyway?
- You repeat some equations in the Appendix (e.g., A.1, A.2) so your equations in the main text link to the equations in the Appendix rather than the corresponding equations in the main text.
- You define what it means for (P,Q) to be tightly dominating for a mechanism but you don't use this terminology anywhere in the paper I think.

---

> ### Author Response · Authors · 2022-12-13
> **answers to reviewer KLwm**
>
> "In my opinion, the title of the paper can be made more indicative of the content."
>
> We agree with this comment and have changed the title.
>
> "In Theorem 6, why wouldn't you just write e^{eps - omega_{P/Q}}? Since you define omega_{P/Q} as a random variable anyway?"
>
> Thank you for the comment, we have made this change.
>
> "You repeat some equations in the Appendix (e.g., A.1, A.2) so your equations in the main text link to the equations in the Appendix rather than the corresponding equations in the main text."
>
> This should be fixed now.
>
> "You define what it means for (P,Q) to be tightly dominating for a mechanism but you don't use this terminology anywhere in the paper I think."
>
> We have removed the definition of 'tightly dominating'.
>
> "What is p in Lemma 11? I don't remember seeing the definition."
>
> We have added to the statement of Lemma 11 reference to def. of P and Q and A and C (p comes from there).
>
> "I believe you could make your analysis even tighter by replacing the Hoeffding confidence intervals in Lemma 11 by tighter confidence intervals for the binomial."
>
> Thank you for the comment. We have decided to leave this for future work as the Hoeffding already enabled evaluating the bounds using laptop.
>
> "Minor: you refer to omega-til as a distribution (for example, in equation (3.5) in Lemma 11) but it's not actually a distribution since you don't renormalize it to add up to 1."
>
> Thank you, we have removed the word 'distribution', it should be correct now.

---

### Review · Reviewer_RsmC · 2022-11-16

**Summary Of Contributions:**

This paper studies privacy amplification by shuffling. The goal is to obtain tighter numerical bounds than prior work, particularly when composing multiple rounds of shuffled mechanisms. They study two main settings:
1. The general setting, where users can use any local randomizer. In this setting, they use the method for analyzing a single round introduced by Feldman et al., and analyze multiple rounds using the FFT technique introduced by Koskela et al.
2. Shuffling k-randomized response. They develop a new upper bound that is based on a weaker adversary than that which appears in Balle et al. Again they use the FFT to analyze the privacy guarantee over multiple rounds.

This paper is well-written and easy to follow. I did not see any technical issues with the paper.

**Audience:**

Yes

**Broader Impact Concerns:**

I don't have any broader impact concerns.

**Claims And Evidence:**

Yes

**Requested Changes:**

- Compare FFT method to performing the composition in RDP (in both the general setting and kRR).
- Compare numerical efficiency improvements for Feldman et al. to the implementation in Feldman et al. git repo.

**Strengths And Weaknesses:**

One of the main theoretical contributions of this paper is the new method for computing the privacy guarantee of kRR (which is new even for a single round). The authors show that this method is tighter than both Balle et al. and Feldman et al. (2021), the two analysis methods available at the time of writing this paper. The authors refer to this analysis as “tight”, although I couldn’t see a formal tightness statement, is it tight for the strong adversary?

In the general numerical comparison, there is a main comparison point missing. The authors use Feldman et al. combined with strong composition as the baseline, although the Rényi amplification version of Feldman et al. (2021) combined with Rényi composition would be a more fair comparison. Feldman et al. show that this analysis beats Girgis et al. for composition over multiple rounds. Without this comparison, I am unconvinced that the FFT method is preferable to RDP composition. Even with this comparison, I'm not sure I see why this result isn't just a simple combination of Feldman et al. and Koskela et al.?

One of the contributions the authors claim in the abstract (and in 3.2) is an improvement in the efficiency of numerically evaluating the method of Feldman et al. I’m confused as to how this compares to efficiency improvements implemented by Feldman et al. in their implementation available on GitHub, and discussed in their appendix. Feldman et al. does not provide a time analysis of their algorithm, but their implementation also involves a speedup that cuts off the tails of the distribution. They also provide another speedup by noticing that when computing the divergence between binomials, the computation simply involves a few cdf computations. The authors mention a 10x speed up for n=10^4 over Feldman et al., what implementation of Feldman et al. is being used for this comparison? (The authors do not cite the GitHub of Feldman et al.)  In Figure 1, the numerical method implemented by the authors performs better than Feldman et al. I was a little confused about this since I thought they were implementations of the same thing (and I would have thought an efficient method would perform slightly worse).

I want to bring the authors attention to Feldman et al.’s follow-up paper “Stronger Privacy Amplification by Shuffling for Rényi and Approximate Differential Privacy” (arxiv: August 2022). Given the timing, I assume this paper was not released when the authors wrote this paper. I think that using the new decompositions from this paper should immediately provide improvements for the results in Section 3. They also provide a numerically tight analysis of kRR, which I suspect would also provide improvements on the results of Section 4.

Minor Comments:
- In the intro, the authors mention that the shuffle model requires LDP mechanisms. This isn’t technically true, Cheu et al. give algorithms that are private in the shuffle model but not built off LDP mechanisms. Similarly, when defining LDP in definition 2, the definition is slightly more nuanced since LDP exists of databases of any size.
- I think there is a typo in Definition 5, t~Q(t)?
- I don’t think A_s is defined before it it is used in Lemma 8.
- Several times the authors reference equations in the appendix. I suspect this is a duplicated labels issue in tex.
- In eqn 4.2, I think n_1 and n_2 should be lower case.
- I appreciated the inclusion of the comparison to Feldman et al. (2021) kRR in the appendix. I would mention the performance in the main body.
- There are two Koskela et al. (2021)s in the appendix and the in text citations do not distinguish between them

---

> ### Author Response · Authors · 2022-12-13
> **answers to reviewer RsmC**
>
> -"when defining LDP in definition 2, the definition is slightly more nuanced since LDP exists of databases of any size"
>
> We have corrected the definition
>
> "I think there is a typo in Definition 5, t~Q(t)?"
>
> For the way we consider the accounting, this is the correct definition. See e.g. Theorem 3.3 of Gopi et al. "Numerical Composition of Differential Privacy" and the definition above it.
>
> "I don’t think A_s is defined before it it is used in Lemma 8."
>
> We have added more details about A_s to the statement of the Lemma.
>
> "Several times the authors reference equations in the appendix. I suspect this is a duplicated labels issue in tex."
>
> Thank you for pointing out, this should be fixed.
>
> "I appreciated the inclusion of the comparison to Feldman et al. (2021) kRR in the appendix. I would mention the performance in the main body."
>
> We have done this.
>
> "In eqn 4.2, I think n_1 and n_2 should be lower case."
>
> We have changed the preceding n_i's upper case so it should be consistent.
>
> "One of the contributions the authors claim in the abstract (and in 3.2) is an improvement in the efficiency of numerically evaluating the method of Feldman et al. "
>
> Thank you for pointing out this. We tried that code and indeed it was a lot faster than our implementation. We were able to make the PLD faster for larger values of eps_0, while for smaller values of eps_0 the RDP accountant seemed to be faster. However, the upper bound given by that method of Feldman et al. seemed to stay loose even when taking tuning the parameters. As there was no clear conclusion to be made, we have left out that statement altogether. In any case, we believe our approach allows computing accurate approx DP bounds for compositions whereas (as such) the numerical method of Feldman et al. does not allow it.

---

> > ### Comment · Reviewer_RsmC · 2022-12-13
> > **Comparison to composition in RDP?**
> >
> > Thank you for your response, in particular to the discussion of the numerical efficiency.
> >
> > My other main concern was regarding a comparison to performing the composition using Feldman et al.'s shuffling bound in RDP, then composition in RDP. Were you able to perform this comparison?

---

> > > ### Author Response · Authors · 2022-12-14
> > > **answer**
> > >
> > > Yes, we computed RDP bounds using the new results by Feldman et al. (2022)  “Stronger Privacy Amplification...", and also for compositions (Figure 1 in the revised pdf). The RDP -> DP conversion is done using formula given by Canonne et al. (2020) ("Discrete Gaussian" paper). The differences are similar as in case of the Gaussian mechanism (zCDP vs. approx DP comparisons in Canonne et al. - paper).
> > >
> > > We need to heuristically speed up the RDP evaluation (n^2 -> n) and we do that using our Lemma 11 which is for the PLD. We believe this could be done rigorously for RDP as well (control the increase of delta).
> > >
> > > Related to your question about tightness of our kRR results: yes they are indeed tight (up to controllable numerical errors), and we believe that would not be straightforward to compute with RDP (at least using existing tools/formulas). So even though those PLD vs. RDP improvements in case of general shufflers (post-processing results of Feldman et al. (2022)) are not that substantial, in case of these kRR evaluations numerical PLD computing seems like a natural choice over RDP.

---

> > > > ### Comment · Reviewer_RsmC · 2022-12-14
> > > > **kRR results**
> > > >
> > > > Thank you for your response and for pointing me towards figure 1 in the revised pdf.
> > > >
> > > > Regarding Fig 4, is the P and Q used from Feldman et al. the ones given specifically for kRR in that paper? Without subsampling, I believe the specific results for kRR given in Feldman et al. should produce the optimal results at least for n_c=1? I'm also a bit confused because the teal lines (FA+clones) look worse in these graphs than they did in Fig 5 in the previous version? It looks like all the parameters have been held constant.
> > > >
> > > > There is still a reference to a speed up in the abstract, is this referring to the speed-up for the computing the Renyi divergence?

---

> > > > > ### Author Response · Authors · 2022-12-20
> > > > > **Answer**
> > > > >
> > > > > Thank you for the comment. We had a closer look at the kRR accountants and ended up finding few errors (which we corrected) :
> > > > >
> > > > > 1) We realized that the bounds we computed for the k-RR shufflers were actually not tight. To compute $\delta$ as a function of $\varepsilon$, we realized we were computing the tail bound of the privacy loss random variable, following the analysis of Balle et al. (2019). We replaced this with the hockey-stick divergence - based numerical accounting which leads to tight bounds (under given adversary assumption).
> > > > >
> > > > > 2) To determine the privacy loss random variable for k-RR under the assumptions of Balle et al. (2019), we realized that one claim by Balle et al. (2019) does not seem to be true: the variables $N_1$ and $N_2$ in the proof of Thm 3.1 of https://arxiv.org/pdf/1903.02837.pdf (page 10) cannot be assumed to be independent binomials, there is some correlation between them. We determined the correct form for the distribution of $(N_1,N_2)$ (certain double binomial, similar to analysis by Feldman et al. (2022)), and as a result obtained the privacy loss random variables for the k-RR shufflers which lead to tight bounds.
> > > > >
> > > > > > I believe the specific results for kRR given in Feldman et al. should produce the optimal results at least for n_c=1?
> > > > >
> > > > > About Figure 5: Now the tight bounds for our 'strong' adversary seem to be very close to the kRR specific bounds by Feldman et al. (2022, thm 5.1). Looking at Thm.5.1 and 5.2 by Feldman et al. (2022) we agree that their kRR specific bound should be tight for $n_c=1$.
> > > > > Notice that our kRR-analysis is based on the assumptions about the adversary considered by Balle et al. (2019, "Privacy blanket"), where the adversary knows the true messages (except for the differing element) and can remove them from the end result. This makes the privacy loss random variables data-independent (i.e. it leads to tight bounds for all neighbouring datasets) which might explain the larger differences when $n_c=4$ and $n_c=16$.
> > > > >
> > > > > > There is still a reference to a speed up in the abstract, is this referring to the speed-up for the computing the Renyi divergence?
> > > > >
> > > > > This $O(n^2)$ to $O(n)$ refers to the speed up we make for the numerical accounting using privacy loss random variables and FFT. E.g., the worst-case distributions $P$ and $Q$ of Feldman et al. (2022) are certain double binomials and thus the PRV will have $O(n^2)$ which is huge already for $n=10^4$. With the Hoeffding bound we can controllably make the $O(n^2)$ to $O(n)$  speed up (have some tolerance parameter for the mass we throw away and add it to $\delta$). We do use this approximation also heuristically for evaluating RDP (it would not have been otherwise possible using laptop).

---

### Review · Reviewer_th4D · 2022-11-29

**Summary Of Contributions:**

This paper aims to give numerical bounds (that is, not closed-form theoretical bounds) on the amplified privacy parameter guaranteed by shuffle differentially private mechanisms, specifically for the settings of composition and subsampling. The authors propose using a method of numerical accounting introduced in prior work, called Fourier Accountant (FA), to calculate a bound on the divergence between two distributions, which measures the privacy loss. The choice of distributions is dictated by other prior work that gives numerical bounds for privacy amplification via shuffling for single mechanisms. This paper extends the latter with the FA method to give numerical bounds for the composition of 2 or more shuffled DP mechanisms and for the case of subsampling without replacement.

**Audience:**

Yes

**Broader Impact Concerns:**

There are no significant ethical implications that would require adding a Broader Impact statement.

**Claims And Evidence:**

No

**Requested Changes:**

The main request is to compare and use the results from the improved analysis of Feldman et al.

**Strengths And Weaknesses:**

Strengths:
I think that the direction of the paper is interesting -- numerical bounds are useful in practice and can be much tighter than closed-form bounds, and the settings of adaptive composition and subsampling are also of practical interest. The paper proposes combining two state-of-the-art methods that give improved numerical bounds for privacy amplification via shuffling and for privacy accounting under composition to provide tight bounds for the both shuffled and composed mechanisms. The experimental evaluations in three different settings (composition of 2-4 mechanisms, subsampling with Renyi DP, and composition of k-RR mechanisms) all seem to show that FA (combined with the right numerical analysis in each case) gives better of the same numerical privacy bounds compared to other approaches.

Weaknesses:
The main concern I have about this paper is that it does not seem to use or compare to the state of the art in these settings. The result on privacy amplification via shuffling by the cited paper of Feldman et al. has been improved in subsequent work: Stronger Privacy Amplification by Shuffling for Rényi and Approximate Differential Privacy (https://arxiv.org/pdf/2208.04591.pdf). This new paper has a better analysis of k-RR mechanisms which is tight numerically (for k>=3) so Balle et al might not be the best comparison in Section 4. It also has an RDP-based analysis which could be useful in the subsampling experiment.

Second, for the experiments on subsampling, it was not very clear to me what the setting was (my understanding was that it is sampling with replacement as in Girgis et al), but more importantly the result imported from Zhu et al. on the comvex combination of the dominating pair of distributions seems to refer to DP under the remove-only neighboring relation between datasets. It would be helpful to clarify this point.

---

> ### Author Response · Authors · 2022-12-13
> **answers to reviewer th4D**
>
> "Weaknesses: The main concern I have about this paper is that it does not seem to use or compare to the state of the art in these settings."
>
> We have updated the P and Q according to the paper mentioned by the reviewer (" Stronger Privacy Amplification by Shuffling for Rényi and Approximate Differential Privacy"). All the formulas for implementation in the appendix have also been updated.
>
> "result imported from Zhu et al. on the comvex combination of the dominating pair of distributions seems to refer to DP under the remove-only neighboring relation between datasets"
>
> As explained in the beginning, we have used Proposition 30 from Appendix of Zhu et al. (2022), and we also found an error in our approach and also a solution via Doroshenko et al. (2022) approach.

---

### Decision · Action_Editors · 2023-02-08

**Recommendation:** Accept with minor revision

**Comment:**

The authors have already conducted a major revision and the current version is significantly improved over the initial submission.
The reviewer feedback and the authors' prompt responses worked out quite well in the process.

I have one more clarification question that potentially requires some minor modifications. The authors wrote:

> We noticed ourselves an error that was not noticed by the reviewers. It was related to the comment of reviewer th4D about the dominating pairs for subsampling:  We were originally using Proposition 30 of Zhu et al. (2022), however we realized that the result does not give a dominating pair (i.e. a pair of distributions that would dominate the alpha divergence for all alpha \geq 0) as the pairs are different for 0 \leq alpha<1 and alpha \geq 1.

The pair is dominating for all alpha > 0 under "remove only" neighbors.  And the other pair is dominating for all alpha > 0 under "add only" neighbors.   What can be done, according to Zhu et al. (2022) is to keep two numerical accountants. One for "remove-only" neighbors the other for "add only" neighbors and take the max only after composition.  This is likely to be tighter than a relaxed bound that tries to get a privacy profile bound (and a dominating pair) for pointwise maximum.

Why is it important to have a more complicated dominating pair for "Add/Remove" together worked out when a tighter alternative with a mere 2x computation is available?

**Audience:**

Theorists and practitioners of differential privacy and differentially private machine learning.

**Claims And Evidence:**

The paper provides a new and refined analysis of the problem of privacy-amplification by shuffling based on a direct analysis of its privacy profile (the curve of $\delta$ as a function of $\epsilon$ in $\epsilon,\delta)$-DP).  The results provide a constant factor improvement over existing approaches based on Renyi DP. The results are especially impressive in the k-RR setting under composition.  The authors also presented a few other technical contributions including handling adversaries with varying degree of capabilities,  and fixing a subtle error from Balle et al. (2019) --- one of the early works that started this line of research.

Overall, the reviewers did a thorough job checking for the correctness of the results. These results are potentially impactful because they will improve the privacy-utility tradeoff in many existing and future implementations of DP (local DP) under the shuffle model.

---

> ### Author Response · Authors · 2023-03-03
> **Answer**
>
> Dear AE,
>
> Thank you, we have submitted a camera ready version and addressed your question by adding a paragraph at the end of Sec. 2.1.
>
> > What can be done, according to Zhu et al. (2022) is to keep two numerical accountants. One for "remove-only" neighbors the other for "add only" neighbors and take the max only after composition.
>
> The reason we do not do this is that Thm. 11 of Zhu et al. (2022) allows doing this if the dominating pair $(P,Q)$ of the underlying mechanism $\mathcal{M}$ is a dominating pair either under remove or add neighbouring relation. However, the post-processing results by Feldman et al. (2023) and our $k$-RR analysis give the dominating pair $(P,Q)$ under the substitute relation. All the results in our paper are stated under the substitute relation.
>
> Instead of using that Thm. 11, we use Thm. 30 given in the appendix of Zhu et al. (2022) that invokes Thm. 9 of Balle et al. (2018, 'Privacy amplification by subsampling: Tight analyses via couplings and divergences') and gives the dominating privacy profile for the subsampled mechanism (w/o replacement) under substitute relation when $(P,Q)$ is a dominating pair under the substitute relation. This is also the setting behind in the baseline results of Girgis et al. (2021) used in our Fig. 3. Then, we need to find a dominating pair for that privacy profile in order to compute tight bounds for compositions. For that we use the method by Doroshenko et al. (2022). Our numerical results (e.g. our Fig. 2) indicate that this construction gives bounds that are tight in a sense that they are very close to the bounds we would obtain, e.g., by using only one of the pairs given by Thm. 30 of Zhu et al. (2022) (that for $\alpha>1$).
>
> Regards,
> The authors

---

> > ### Comment · Action_Editors · 2023-03-07
> > **Thanks for the notes**
> >
> > That clarifies matters! Thanks a lot and congratulations on a nice paper!